# Purine Nucleotide Alterations in Tumoral Cell Lines Maintained with Physiological Levels of Folic Acid

**DOI:** 10.3390/ijms241612573

**Published:** 2023-08-08

**Authors:** Claudia Cano-Estrada, Lidia de Benito-Gómez, Paula Escudero-Ferruz, Neus Ontiveros, Daniel Iglesias-Serret, José M. López

**Affiliations:** 1Institut de Neurociències, Universitat Autònoma de Barcelona, 08193 Cerdanyola del Vallès, Barcelona, Spain; 2Departament de Bioquímica i Biologia Molecular, Unitat de Bioquímica, Facultat de Medicina, Universitat Autònoma de Barcelona, 08193 Cerdanyola del Vallès, Barcelona, Spain; 3Departament de Ciències Fisiològiques, Facultat de Medicina i Ciències de la Salut, Universitat de Barcelona-IDIBELL (Institut d’Investigació Biomèdica de Bellvitge), 08907 L’Hospitalet de Llobregat, Barcelona, Spain; 4Facultat de Medicina, Universitat de Vic-Universitat Central de Catalunya (UVic-UCC), 08500 Vic, Barcelona, Spain

**Keywords:** purines, nucleotides, ZMP, AICAR, folic acid, SLC19A1, cancer

## Abstract

Most cancer cells have an increased synthesis of purine nucleotides to fulfil their enhanced division rate. The de novo synthesis of purines requires folic acid in the form of N^10^-formyltetrahydrofolate (10-formyl-THF). However, regular cell culture media contain very high, non-physiological concentrations of folic acid, which may have an impact on cell metabolism. Using cell culture media with physiological levels of folic acid (25 nM), we uncover purine alterations in several human cell lines. HEK293T, Jurkat, and A549 cells accumulate 5′-aminoimidazole-4-carboxamide ribonucleotide (ZMP), an intermediary of the de novo biosynthetic pathway, at physiological levels of folic acid, but not with the artificially high levels (2200 nM) present in regular media. Interestingly, HEK293T and Jurkat cells do not accumulate high levels of ZMP when AICAr, the precursor of ZMP, is added to medium containing 2200 nM folate; instead, ATP levels are increased, suggesting an enhanced de novo synthesis. On the other hand, HeLa and EHEB cells do not accumulate ZMP at physiological levels of folic acid, but they do accumulate in medium containing AICAr plus 2200 nM folate. Expression of SLC19A1, which encodes the reduced folate carrier (RFC), is increased in HEK293T and Jurkat cells compared with HeLa and EHEB, and it is correlated with the total purine nucleotide content at high levels of folic acid or with ZMP accumulation at physiological levels of folic acid. In conclusion, tumoral cell lines show a heterogenous response to folate changes in the media, some of them accumulating ZMP at physiological levels of folic acid. Further research is needed to clarify the ZMP downstream targets and their impact on cell function.

## 1. Introduction

The synthesis of nucleotides is essential for cells as a source of energetic and structural requirements. Both purines and pyrimidines are needed for cellular survival and proliferation. There are two different pathways for the synthesis of purines in eukaryotic cells: the de novo and the salvage pathway [1].

The de novo biosynthetic pathway has a very high energy demand. Just to synthesize one IMP, several molecules are needed: four ATP, two glutamines, two formats as 10-formyl-THF and one molecule of each glycine, aspartate, and carbon dioxide. Six different enzymes catalyze the ten steps in this route to convert 5-phosphoribosyl-1-pyrophosphate (PRPP) into IMP [1]. Interestingly, a network organization of these enzymes in the cytosol forms a macromolecular complex, named purinosome, which is in close contact with the mitochondria [2]. As expected for an expensive metabolic pathway, purine synthesis is regulated at the first step, in a reaction catalyzed by phosphoribosyl pyrophosphate amidotransferase (PPAT), through feedback inhibition by end products, purine ribonucleotides, obtained either by the de novo or salvage pathways [3]. The salvage pathway, which is energetically cheaper, is regulated by the enzymes hypoxanthine-guanine phosphoribosyltransferase (HGPRT) and adenine phosphoribosyltransferase (APRT), which form purine nucleotides in one step reaction by using purine bases and PRPP as substrates (Figure 1A). The complete deficiency of HGPRT activity causes Lesch–Nyhan disease (LND). Patients present severe neurological problems such as spasticity, cognitive deficit, dystonia, and self-injurious behavior. At a biochemical level, LND patients have hyperuricemia because hypoxanthine and guanine cannot be recycled, and they are degraded to uric acid instead [4,5]. Moreover, there is an increased acceleration of de novo purine synthesis, which tries to compensate for the deficient salvage pathway, further increasing the production of uric acid [6,7]. Recently, we have described the accumulation of 5-aminoimidazole-4-carboxamide riboside 5′-monophosphate (ZMP, also named AICAR), an intermediary of the de novo purine biosynthetic pathway, in LND fibroblasts maintained with physiological levels of folic acid (FA) [8]. This alteration likely results from the fact that conversion of ZMP to IMP by the enzyme 5-aminoimidazole-4-carboxamide ribonucleotide formyltransferase/IMP cyclohydrolase (ATIC) is folate dependent (Figure 1A). A derivative of ZMP, the nucleoside AICAr, is present in the urine and the cerebrospinal fluid of patients with LND, but not in control individuals, suggesting that a similar alteration may occur in vivo [8].

Folic acid, in the form of 10-formyl-tetrahydrofolate (THF), mediates the transfer of 1C units in de novo purine synthesis in two different reactions: the conversion of GAR to FGAR catalyzed by 5′-phosphoribosyl-glycinamide transformylase (GART) and the conversion of ZMP to FAICAR catalyzed by ATIC (Figure 1A). Although both GART and ATIC are folate dependent, ATIC seems to limit the rate of folic acid depletion [9] for reasons that are still not clear. To obtain 10-formyl-THF, multiple reactions are involved in creating a network known as folate-mediated one-carbon metabolism (FOCM), which is compartmentalized inside the mitochondria and the cytosol (Figure 1B) [10]. It has been proposed that the mitochondrial folic acid cycle is the main contributor of 1C units for nucleotide biosynthesis in cancer cells [11]. However, recently, Lee et al., have reported that human cell lines expressing low levels of the reduced folate carrier (RFC) SLC19A1 preferentially use the cytosolic versus mitochondrial one-carbon flux when the cells are maintained with physiological levels of FA [12]. Because tumoral cells usually have an accelerated de novo purine biosynthesis to support a high proliferation rate [13,14,15,16,17], we assessed whether cancer cells maintained with physiological levels of FA, which are not deficient in HGPRT, also accumulate ZMP. Here, we show that Jurkat cells mimic the purine alterations previously reported in LND fibroblasts [8]. Moreover, ZMP accumulation was also present in A549 and HEK293T cell lines, but not in HeLa or EHEB, thus indicating differential responses of the cells to physiological levels of FA. Interestingly, the cell lines that accumulate ZMP have higher levels of the reduced folate carrier (RFC) SLC19A1 and an efficient metabolization of AICAr, the precursor of ZMP, in regular media with high folate levels.

## 2. Results

### 2.1. Physiological Levels of FA Induce ZMP Accumulation in Jurkat Cells

Jurkat is a human cell line established from the peripheral blood of a patient with acute T-cell leukemia [18]. ATCC recommends culturing these cells with RPMI medium at a concentration between 1 × 10^5^ and 1 × 10^6^ cells/mL, not allowing the cell density to exceed 3 × 10^6^ cells/mL [19]. One of the most fundamental parameters that any healthy cell must maintain is a high ratio of ATP to ADP, as well as a high ratio of ADP to AMP [20]. In preliminary experiments, we found that Jurkat cells maintained with regular RPMI at a high density (2 × 10^6^ or 1.5 × 10^6^ cells/mL) presented higher levels of AMP than ATP, as well as higher levels of GMP than GTP. When the culture was maintained at a density of 5 × 10^5^ cells/mL, the ATP/ADP/AMP and GTP/GDP/GMP ratios were well preserved, as expected for healthy cells (Appendix A). Therefore, all the experiments with Jurkat cells were done at low density (5 × 10^5^ cells/mL or below) to preserve an optimal purine nucleotide ratio.

Recently, we have explained that human fibroblasts from LND patients accumulate ZMP when the cells are cultured with RPMI containing physiological levels of FA (25 nM), instead of the nonphysiologically high levels present in regular medium (2200 nM) [8]. We addressed whether Z-metabolites accumulate in Jurkat cells maintained in these conditions. Cells were cultured during 5 days with RPMI medium supplemented either with 2200 nM FA or 25 nM FA, and perchloric acid (PCA) extracts were obtained to determine Z-metabolites using the Bratton-Marshall test. Significantly higher levels of Z-metabolites were detected in cells cultured with physiological levels of FA (25 nM) compared with 2200 nM FA (Figure 2A). However, the Bratton-Marshall test cannot distinguish between ZMP and other derivatives (AICA, AICAr, ZDP, ZTP) and could even detect other precursors in the de novo purine biosynthetic pathway, like AIR, CAIR or SAICAR [21]. Therefore, PCA extracts were analyzed using high-performance liquid chromatography (HPLC). As shown in Figure 2B, high levels of ZMP were detected in the cells maintained at 25 nM FA but were almost undetectable at 2200 nM FA.

Cantor et al.; reported that medium containing uric acid (UA) at a physiological concentration (350 µM) inhibits uridine monophosphate synthase (UMPS) thus altering pyrimidine, but not purine, biosynthesis [22]. As shown in Figure 2B, Jurkat cells maintained in RPMI containing 350 µM UA and 25 nM FA (both physiological concentrations) presented similar levels of ZMP to cells maintained with only 25 nM FA. In conclusion, physiological levels of folic acid induce ZMP accumulation in Jurkat cells independently of the presence of uric acid in the medium. Similar results were obtained when the data were expressed in µM (Appendix A).

### 2.2. Physiological Levels of FA do Not Induce AMPK Activation in Jurkat Cells

Intracellular levels of the most abundant purine nucleotides were determined by HPLC (Table 1). When the results were expressed as nmol/mg protein, there were not significant differences in the levels of ATP, ADP, AMP, GTP, GDP, and GMP in the cells treated with different media. There was a tendency for lower ATP levels in cells maintained with physiological FA (25 nM), that only reached statistical significance when the results were expressed as µM (Appendix A). The AMP/ATP ratio did not change with the different media (Figure 3A), as well as the adenylate energy charge (Figure 3B), which is defined as the ratio ([ATP] + 1/2 [ADP])/([ATP] + [ADP] + [AMP]) by Atkinson and Walton [23]. AMPK was not activated in Jurkat cells maintained with physiological levels of FA or physiological levels of FA plus UA (Figure 3C). The pAMPK/AMPK ratio did not change in the cells maintained with different media, when analyzing either the cytoplasmic or total extracts (Figure 3D).

### 2.3. Physiological Levels of FA Decrease Tetrazolium Dye Reduction by Jurkat Cells without Changes in Mitochondrial Membrane Potential

Next, we measured the capacity of the reduction of tetrazolium dye (MTT assay) by Jurkat cells cultured for 5 days in different media. A significant decrease in tetrazolium dye reduction was observed in cells maintained with physiological levels of FA, either with or without the presence of UA, compared with cells maintained in medium containing high levels of FA (Figure 4A). No significant differences were obtained in the doubling time of the cells (Figure 4B). Moreover, Jurkat cells maintained with different media did not present any change in mitochondrial membrane potential, measured as DilC signal by cell cytometry (Figure 4C). As a positive control of mitochondrial depolarization, cells were incubated with the disrupter CCCP. Lee et al.; have reported that Jurkat cells maintained with physiological levels of FA have lower levels of mitochondrial NADPH [12]. This could explain the decreased activity detected in the MTT assay (see Discussion).

### 2.4. ZMP Levels in Different Human Cell Lines Maintained with Physiological Folate

Next, we addressed whether ZMP accumulation detected in Jurkat cells also occurred in other human cell lines. As shown in Figure 5A, significant levels of ZMP, expressed as nmol/mg protein, were detected in HEK293T and A549 cell lines maintained with RPMI containing 25 nM FA, but not in HeLa and EHEB cells. HEK293T accumulated higher levels of ZMP than Jurkat and A549 and, accordingly, ZTP was detected only in HEK293T cells maintained with physiological folate (Figure 5D). None of the cell lines analyzed showed accumulation of ZMP or ZTP when maintained with RPMI containing 2200 nM FA (Figure 5A,D). ATP levels were decreased in HEK293T cells maintained with physiological FA (Figure 5B). No significant differences were observed for other nucleotides comparing high FA condition with physiological FA within each cell line (Figure 5C,E,F,H,I).

The doubling time of HEK293T, Jurkat, and A549 cells maintained with 2200 nM FA was similar (22–23 h), higher in HeLa (29 h), and very high in EHEB cells (63 h) (Figure 5G). The differences between EHEB and the other cell lines were statistically significant. The total content of purine nucleotides (adenylates plus guanylates) in cells maintained with 2200 nM FA was higher in HEK293T, Jurkat, and A549 than in HeLa and EHEB (Figure 5J, dotted bars). The difference between HEK293T and HeLa was statistically significant. Cell lines maintained with physiological levels of FA presented a similar content of purines (Figure 5J, solid bars). Note that HEK293T, Jurkat, and A549 cell lines decreased the total purine content at 25 nM FA compared to 2200 nM FA, being statistically significant in HEK293T (Figure 5J).

In conclusion, physiological levels of FA induce ZMP accumulation in some tumoral cell lines (HEK293T, Jurkat, A549) but not in others (HeLa, EHEB).

### 2.5. Differential Effect of AICAr in Purine Nucleotide Levels of Tumoral Cell Lines

ZMP could accumulate in cells if there is increased de novo purine biosynthesis and limited availability of folic acid. To assess whether the differential accumulation of ZMP between cell lines was due to a different flux of de novo purine biosynthetic pathway, cells were incubated with RPMI containing 2200 nM FA with or without 2 mM AICAr for 24 h. AICAr is a nucleoside that, after incorporation into the cells, is phosphorylated to ZMP (AICAR) by adenosine kinase [24]. Therefore, cells with an accelerated de novo purine synthesis and high levels of FA should metabolize ZMP into ATP and/or GTP. As shown in Figure 6A, HEK293T cells treated with AICAr did not accumulate ZMP and Jurkat cells only did so at very low levels, but both presented higher levels of ATP compared with non-treated cells (Figure 6B). Instead, HeLa and EHEB cells treated with AICAr accumulated high levels of ZMP (Figure 6A), without any significant change in the ATP pool (Figure 6B). ZTP was detected in the cells that accumulated high levels of ZMP (Figure 6D). A459 cells showed an intermediate response, accumulating higher levels of ZMP than Jurkat but lower than HeLa and EHEB (Figure 6A). In conclusion, ZMP accumulation in the cell lines treated with 2 mM AICAr plus 2200 nM FA (Figure 6A) was inversely correlated with ZMP accumulation at physiological levels of FA (Figure 5A). GTP, GDP, GMP, ADP and AMP levels did not change significantly with AICAr treatment (Figure 6C,E,F,H,I). AICAr treatment increased the total purine content of HEK293T, Jurkat and A549 cell lines, but reduced the purine levels of HeLa and EHEB (Figure 6G). AICAr treatment also decreased the cell growth, between 30 and 60% in all the cell lines (Appendix A). It is well known that AICAr inhibits cell cycle and cell growth in an AMPK-independent manner [25,26,27]. Both biological effects, the increase in purines and cell growth inhibition, can occur simultaneously in cells treated with AICAr [27].

HeLa and EHEB cells treated with AICAr plus 2200 nM folic acid accumulate ZMP, contrary to HEK293T cells. This could be explained by an increased uptake of AICAr and/or phosphorylation into ZMP by adenosine kinase in HeLa and EHEB cells. However, intracellular levels of AICAr (Appendix A) and cytosolic adenosine kinase expression (Appendix A) were similar in all the cell lines except Jurkat, which expressed lower levels of adenosine kinase and significantly higher levels of intracellular AICAr. Therefore, ZMP accumulation induced by AICAr treatment (Figure 6A) is not correlated with AICAr intracellular levels or adenosine kinase expression. Altogether, the previous results suggest that HEK293T cells preferentially use the de novo purine biosynthetic pathway to fulfil their purine nucleotide requirements, as opposed to HeLa or EHEB. However, we should be cautious with this conclusion because we have not measured nucleotidases or purine recycling in the different cell lines, and this is certainly a limitation of our work. More studies are necessary to know the relative fluxes of the de novo and salvage pathways, as well as the nucleotidase activity, in the different cell lines.

### 2.6. Expression of the Reduced Folate Carrier (RFC) SLC19A1 in Human Cell Lines

In the de novo purine synthesis, folic acid, in the form of tetrahydrofolate (THF), is the carrier of 1C units primarily derived from amino acid catabolism. THF is converted to formyl-THF through multiple reactions taking place in the cytosol and/or the mitochondria (Figure 1B). Because folic acid is a vitamin that cannot be synthesized inside the cells, transport from the extracellular environment is essential. There are three types of folate transporters: the reduced folate carrier (RFC), proton-coupled folate transporter (PCFT), and folate receptors (FRs) [28]. Recently, it has been reported that human cell lines expressing low levels of SLC19A1 preferentially use the cytosolic carbon flux, instead of the mitochondrial, for the synthesis of nucleotides [12].

To assess whether the differential accumulation of ZMP detected in our cell lines (Figure 5A and Figure 6A) was correlated with SLC19A1, we measured the mRNA expression levels of the folate transporter by qRT-PCR in cells maintained with physiological levels of FA (25 nM), high levels of FA (2200 nM), or with AICAr (2 mM) plus high levels of FA (2200 nM). As shown in Figure 7A,B, HEK293T cells expressed higher levels of SCL19A1 than HeLa and EHEB, independently of the media used for cell culture (Figure 7B). Jurkat and A549 cell lines expressed intermediate levels. Therefore, the rank of expression of SCL19A1 in the cell lines was directly correlated with the levels of ZMP at physiological levels of FA (r = 0.94) (Figure 7C) and inversely correlated with ZMP levels when the cells were incubated with AICAr plus 2200 FA (r = −0.96) (Figure 7D). The expression of SLC19A1 was not affected by FA concentration in the media (Figure 7A), as previously reported [12]. There was a positive correlation between total purine content and SCL19A1 expression when the cell lines were maintained with 2200 nM FA, with or without AICAr (r = 0.89) (Figure 7E,G). This correlation was completely lost in cells maintained with 25 nM FA because the differences in total purine content between cells lines were reduced (Figure 7F).

To conclude, tumoral cells expressing high levels of SCL19A1 increase their total purine content when maintained at high levels of folate but accumulate ZMP at physiological levels of folic acid.

## 3. Discussion

### 3.1. Jurkat Cells Maintained with Physiological Folate Accumulate ZMP, but AMPK Is Not Activated

Jurkat cells accumulate ZMP under physiological levels of folate. An increase in ZMP could, in principle, activate AMPK. However, as we show in Figure 3G,H, AMPK is not activated at physiological levels of FA. It is well known that ZMP is around 50-fold less potent than AMP in inducing AMPK activation [29]. AMPK will only be activated when ZMP reaches millimolar concentrations and ATP is significantly reduced. In fact, we have reported that Jurkat cells treated with AICAr for 24 h moderately increase ZMP levels and do not activate AMPK [30]. The AMP/ATP ratio does not change in cells maintained with physiological levels of FA (Figure 3A), thus explaining why AMPK is not activated.

Interestingly, Jurkat cells maintained with physiological folate showed lower tetrazolium dye reduction (MTT assay), suggesting a metabolic alteration in the mitochondria. It has been reported that ZMP, at high concentrations, inhibits respiration in purified mitochondria through a direct effect on the respiratory-chain complex I [31]. However, Jurkat cells maintained with physiological FA did not present any change in the mitochondrial membrane potential, measured as DilC signal by cytometry (Figure 4C), making a direct effect of ZMP in the mitochondria unlikely. It has been reported that Jurkat cells maintained at physiological levels of FA mainly rely on the cytosolic 1C flux, instead of the mitochondrial, thus decreasing the contribution of serine catabolism to a reduction in mitochondrial NADP^+^ via MTHFD2 [12]. A drop in mitochondrial NADPH could explain the decreased activity measured by the MTT assay. However, although it is assumed that tetrazolium salt reduction is related to energy metabolism, most reduction appears to be non-mitochondrial [32]. More studies are necessary to clarify why the activity measured by the MTT assay is folate dependent and whether NADPH or ZMP could be related.

Patients with Lesch–Nyhan disease (LND) present megaloblastic anemia [33]. Alterations in B and T lymphocytes have also been reported in LND [34,35], although more studies are necessary. The scarce number of patients and the reduced number of lymphocytes obtained from blood samples are limitations for biochemical research. Because Jurkat cells reproduce the purine alterations observed in LND fibroblasts [8], we could use these cells as a cellular model to study downstream alterations and potential targets involved in the pathology of LND. They could be used, for instance, to find drugs that block ZMP accumulation.

### 3.2. Differential ZMP Accumulation in Human Cell Lines

Here, we show that other human cell lines (HEK293T, A549) commonly used in biomedical research also accumulate significant levels of ZMP when they are maintained with RPMI containing physiological levels of FA (25 nM), reaching values very similar or even higher (HEK293T) than intracellular AMP. When these cells are maintained with RPMI containing 2200 nM FA, an artificially high level present in almost all commercial media, ZMP is undetectable by HPLC. However, this behavior is not observed in all the human cell lines analyzed. HeLa and EHEB cells do not accumulate ZMP with physiological folate (Figure 5A). EHEB is a human cell line established from the peripheral blood of a patient with B cell-chronic lymphocytic leukemia (B-CLL) [36]. The doubling time of EHEB cells is much longer than in the other cell lines (Figure 5G), implying a low demand for nucleotide biosynthesis. Therefore, the de novo purine biosynthetic flux and ZMP levels are expected to be low in this cell line. However, HeLa cells have a similar doubling time to Jurkat and A549 cells at physiological levels of FA, but they do not accumulate ZMP. These results suggest that HeLa and EHEB are less dependent on folic acid supply to fulfil their purine nucleotides requirements. Interestingly, when these cell lines are incubated with 2 mM AICAr plus 2200 nM FA, they accumulate high levels of ZMP, whereas HEK293T does not (Figure 6A), suggesting that ZMP is efficiently metabolized through the de novo purine pathway in HEK293T cells. AICAr intracellular concentration and adenosine kinase expression are similar in HEK293T and HeLa (Appendix A), thus ruling out a differential incorporation and/or phosphorylation of AICAr in these cell lines. More studies are necessary to know the relative fluxes of the de novo and salvage purine nucleotide biosynthetic pathways. In addition, it would be interesting to investigate whether the differential ZMP accumulation obtained between cell lines at physiological levels of folic acid can also be reproduced with 5-methyltetrahydrofolate, which is the is the main folate in plasma [10].

### 3.3. Folate Transport and ZMP Accumulation

Because of their aberrant proliferative behavior, one-carbon metabolism is especially important in cancer cells. The mitochondrial folic acid cycle and formate export from the mitochondria to the cytosol are the main contributors of 1C units for nucleotide biosynthesis in cancer cells maintained in regular medium containing high, non-physiological concentrations of FA [11]. However, in 50% of cancer cell lines maintained with physiological levels of FA, the cytosolic pathway seems to be the predominant source of 1C units. Low levels of the reduced folate carrier SLC19A1 in these cell lines favor the cytosolic versus the mitochondrial pathway [12].

Our results show that high expression of SLC19A1 in human cell lines correlates with accumulation of ZMP at physiological levels of FA (Figure 7C), suggesting that these cells need high amounts of 10-formyl-THF for purine biosynthesis. The differential expression of SLC19A1 between cell lines obtained by qRT-PCR was similar to the transcriptomic data reported in the Cell Line Encyclopedia Collection (CCLE) [37] for four cell lines analyzed (data for HEK293T were not available in the CCLE). High expression of SLC19A1 is correlated with high purine content when cells are maintained in a medium with 2200 nM FA (Figure 7E). Without folic acid restrictions, these cells are well suited to cell proliferation. At physiological levels of FA, however, ZMP accumulates, acting as a sensor of FA depletion. We still do not know the downstream targets regulated by ZMP.

Although we have not measured fluxes of folate 1C metabolism and pyrimidines, we hypothesize that, at physiological levels of folic acid, cells that express high levels of SLC19A1 will preferentially use the de novo purine biosynthetic pathway and the mitochondrial folate cycle as a source of 1C units for the synthesis of both purines and pyrimidines, whereas cells that express low levels of SLC19A1 would rely on the salvage pathway for purines and on the cytosolic folate pathway for pyrimidines. It would be interesting to test this hypothesis by analyzing metabolic fluxes. Purinosome formation has been detected in HGPRT deficient cells [7] and in cancer cells maintained with low levels of purines [2]. The role of folate levels on purinosome formation deserves further investigation.

### 3.4. Could ZMP, or Any of Its Derivatives, Be Used as a Tumoral Marker?

Cancer cells present an increased de novo purine biosynthesis to fulfil their high nucleotide demands [13,14,15,16,17]. Oncogene activation and loss of tumor suppressors reprogram nucleotide metabolism in cancer cells, increasing de novo purine synthesis [16]. Some enzymes in this pathway (PPAT, PAICS) are expressed at higher levels in specific types of cancer [38,39]. However, under normal physiological conditions, the expression levels of the enzymes of the de novo pathway usually remain unaltered [40,41]. For instance, in purine depleted medium, clustering of the enzymes that form the purinosome increases de novo purine synthesis [40]. Moreover, physiological and oncogenic ERK signaling activation stimulates the de novo purine synthesis through direct phosphorylation of the purine enzyme FGAMS [41]. In addition, it has been reported that media folate level has no effect on the concentration of key 1C metabolic enzymes [12].

In 2012, Keller et al.; reported that SAICAR, the precursor of ZMP in the de novo pathway, accumulates in several human cancer cell lines upon glucose depletion [42]. However, only a small portion of cancer cells might suffer glucose deprivation in vivo, making its use as a tumoral marker unlikely.

As we have found that ZMP is accumulated in several human cell lines with high proliferative rates (HEK293T, Jurkat, A549), one could ask whether ZMP can be used as a tumoral marker. As we show here, this is not a general phenomenon because HeLa cells have a high proliferative rate but do not accumulate ZMP. However, it is worth wondering if this nucleotide, or its derivatives, could be detected in the blood or urine of some types of cancer patients. ZMP and its derivatives (AICAr and AICA) are easily detected in biological samples by using the Bratton-Marshall test, which is a colorimetric method [8,43]. ZMP accumulates in cancer cells treated with pemetrexed, an anti-folate drug, because the enzyme ATIC is a target of antifolates [9]. Moreover, patients treated with methotrexate present increased levels of AICA in urine [44]. Additional studies are required to assess whether tumoral cells of patients not treated with antifolates accumulate ZMP in vivo to produce significantly high levels of AICAr and/or AICA in the blood or urine. Until now, high levels of AICAr and AICA have been detected only in patients with an altered purine metabolism, like ATIC or HGPRT deficiencies [8,45].

## 4. Materials and Methods

### 4.1. Cell Culture

HEK293T (Cultek HCL4517), HeLa (ECACC 93021013), A549 (ECACC 86012804), Jurkat (ECACC 88042803) and EHEB (DSMZ 14916) cells were provided by the culture service of the INc-UAB. Cells were maintained in RPMI without folic acid (Sigma-Aldrich, Burlington, MA, USA, R1145) supplemented with 10% fetal bovine serum (FBS) (Sigma-Aldrich, T7524) (except EHEB, supplemented with 20% FBS), 2200 nM filtered FA (Sigma-Aldrich, F8758), 1% L-glutamine (Gibco, Waltham, MA, USA, 250381), 1% Pen/Strep (Gibco, 250381) and 0.2% sodium bicarbonate (Sigma-Aldrich, S8761). The day before treatment, 160,000 adherent cells (HEK293T, HeLa, A549) were seeded in 100 mm dishes. Jurkat and EHEB cells were seeded on the first day of the treatment at a concentration of 100,000 cells/mL and 350,000 cells/mL, respectively. Before the treatment, dishes were washed several times with PBS and then maintained for 5 days in RPMI medium with 2200 or 25 nM FA. Some cells were maintained for 4 days in RPMI with 2200 nM FA and AICAr (Toronto Research Chemicals, North York, ON, USA, A611700) was added to this medium, at a final concentration of 2 mM, for 24 h.

### 4.2. Cell Growth

Cells were counted at 2 day intervals using a Neubauer chamber after trypan blue staining. The population doubling time was estimated by applying the following formula (which considers a linear regression during the exponential growth phase):$$\mathrm{Doubling}\;\mathrm{time}=\;\frac{\mathrm{Ln}\;2\;\times \;\mathrm{Time}\;\left(\mathrm{hours}\right)}{\mathrm{Ln}\;\frac{\mathrm{Final}\;\mathrm{cell}\;\mathrm{concentration}}{\mathrm{Initial}\;\mathrm{cell}\;\mathrm{concentration}}}$$

### 4.3. Purine Measurement

For nucleotide extracts, cells were harvested by trypsinization (HEK293T, HeLa and A549) or centrifugation (Jurkat and EHEB), counted in a Neubauer chamber and resuspended in 0.4 N perchloric acid (PCA). After 15 min on ice, cells were centrifuged at 12,000× *g* for 5 min at 4 °C. Pellet was stored at −20 °C for later protein quantification and supernatant was neutralized with 5 M potassium carbonate (Sigma-Aldrich, 20961) and filtered through PVDF micro-spin filters (Thermo Scientific, Waltham, MA, USA, F2517-5) via centrifugation at 10,000× *g* for 10 min at 4 °C.

Two different methods were used to determine purines.

The first method, the Bratton-Marshall test [46], modified with prior addition of acetic anhydride for the acetylation of amines [47], was used to quantify Z-metabolites. Briefly, in a 96-multiwell plate, 2 μL of acetic anhydride (Sigma-Aldrich, 320102) was added to 50 μL of perchloric extract (1:2 dilution in mQ water) and incubated for 30 min at room temperature. Then, 10 μL of hydrochloric acid 4N (AnalaR) and sodium nitrite 0.1% (Sigma-Aldrich, 237213) were added and incubated for 3 min on ice. Next, 10 μL of ammonium sulfamate 0.5% (Sigma-Aldrich, 237213) was added and incubated on ice for 2 extra min. At this point, absorbance at 540 nm was measured and the value obtained was used as a blank for each sample. Finally, 10 μL of naphthylethylenediamine dihydrochloride (NEDA) 0.1% (Sigma-Aldrich, N9125) was added and after 5 min on ice, and a new measure of absorbance at 540 nm was taken. A standard curve of increasing concentrations of ZMP was prepared (0, 2.5, 5, 10 and 20 μM) and used for quantification.

The second method involved using high pressure liquid chromatography (HPLC) coupled with a UV detector. Analytes were separated using reverse-phase ion-pair chromatography on an Atlantis T3 column (Waters, 186003729). The optimized method for nucleotides’ separation and quantification consists of a sequence of stepped gradients of buffer A (10 mM of ammonium acetate (Sigma-Aldrich, A1542) and 2 mM of tetrabutylammonium phosphate monobasic solution (Sigma-Aldrich, 268100), pH5) and buffer B (10 mM of ammonium phosphate (Sigma-Aldrich, 09709), 2 mM of tetrabutylammonium phosphate, and 25% of acetonitrile (J.T. Baker, 76045) pH7). The gradient sequence includes the following steps: 100% of buffer A for 10 min, a linear gradient of buffer B up to 75% over 10 min, 9 min at 75% buffer B, a linear gradient to 100% buffer B over 3 min, 8 min at 100% buffer B, a linear gradient to 100% A in 1 min, and finally, 10 min maintained at 100% buffer A. The identification of purines was carried out by comparing their retention times to known standards and they were quantified at 254 nm by using standard curves of all the compounds analyzed. A sample analysis was performed using EZ Chrom Elite/ELITE UV-VIS 3.1.7 software.

To determine the intracellular concentration of nucleotides, the volume of Jurkat cells (1.00 × 10^−6^ μL) was empirically calculated by measuring the cell diameter and considering its spherical shape.

### 4.4. Protein Quantification

Protein quantification was determined by using a commercial Micro BCA Protein Assay Kit (Pierce, Waltham, MA, USA, 23235). Protein extracts were resuspended in 2% SDS and incubated overnight at 37 °C to completely dissolve the protein pellet. 1:100 dilutions were produced and incubated with reagent solution at 37 °C for 2 h. After that, absorbance was measured at 625 nm in a PowerWave XS spectrophotometer (Bio-Tek, La Puenta, CA, USA). A standard curve with increasing concentrations of BSA was prepared and used for quantification.

### 4.5. MTT Assay

Jurkat cells, at a concentration of 5 × 10^5^ cells/mL, were maintained for 5 days in different media. Next, 100 μL of each cell culture was trespassed to a 96-multiwell plate and incubated with 0.45 mg/mL of 3-(4,5-dimethyl-2-thiazolyl)-2,5-diphenyl-2H-tetrazolium bromide (MTT) for 2 h at 37 °C. Violet crystals were observed using microscopy and 100 μL of isopropanol:HCl (24:1) was added by pipetting up and down to lysate the cells. Finally, absorbance at 550 nm was measured in a PowerWave XS spectrophotometer (Bio-Tek).

### 4.6. Mitoprobe DilC_1_(5) Assay

This assay was performed following the instructions detailed in the MitoProbe DilC_1_(5) Assay kit for Flow Cytometry (Invitrogen, Carlsbad, CA, USA, M34151). For each sample, 3 × 10^5^ cells were resuspended in 500 µL of phosphate-buffered saline (PBS). As a control of mitochondrial depolarization, samples were incubated for 5 min with 50 µM of CCCP, final concentration, at 37 °C. Then, DilC was added to achieve a final concentration of 10 nM and incubated for 30 min at 37 °C, 5% CO_2_. Cells were washed and resuspended in 300 µL of PBS. Each tube was analyzed using flow cytometry with 633 nm excitation and a suitable filter for Alexa Fluor.

### 4.7. Western Blot

Total cell lysate was obtained by the addition of 250 µL lysis buffer A (50 mM Tris, 2% SDS pH 6.8) to 2 × 10^6^ cells and heating at 100 °C for 5 min. For the cytosolic fraction, cells were lysed by using lysis buffer B (270 mM sucrose, 1 mM EDTA, 1mM EGTA, 1% Triton x-100, 1 mM DTT, 50 mM Tris pH 7.5, protease inhibitors: 0.1 mg/mL leupeptin, 0.01 mg/mL aprotinin, 1 mM PMSF; and phosphatase inhibitors: 50 mM NaF, 5 mM sodium pyrophosphate, 10 mM βGP, 1 mM sodium orthovanadate). Then, the extracts were centrifuged at 13.000 rpm for 15 min at 4 °C, and the supernatant was collected as cytosolic fraction. Both total and cytosolic extracts were preserved at −20 °C.

Samples collected (40 µg protein) were denatured in Laemmli sample buffer at 100 °C for 5 min. Electrophoretic separation was performed in a 12% polyacrylamide gel and proteins were transferred to a PVDF membrane in transfer buffer (25 mM Tris, 192 mM glycine, 10% methanol) at 400 mA for 2 h. After transference, the membrane was washed with TBST (25 mM Tris, 135 mM NaCl, pH 7.5 and 0.1% Tween20) and blocked for 90 min with 5% BSA (Sigma-Aldrich, A9647) in TBST, washed several times with TBST, and then incubated overnight at 4 °C with primary antibodies for AMPK (Cell Signaling, 2532) or pAMPK (Cell Signaling, 2531) diluted at 1:1000 in TBST containing 5% BSA. Antibody binding was detected next morning by washing the membrane with TBST several times and incubating for 2 h at room temperature with horseradish peroxidase-coupled (HRP) anti-rabbit secondary antibody (Invitrogen, 31460).

The membrane was washed several times with TBST, and the signal was detected using chemiluminescence in Biorad Chemidoc. Resulting bands were analyzed and quantified using the Image-J program.

### 4.8. qRT-PCR

Total RNA was extracted from the different cell lines using the Maxwell RSC simplyRNA Cells Kit (Promega, Madison, WI, USA, AS1390) following the manufacturer’s instructions. This procedure includes DNase I treatment to prevent DNA contamination. Reverse transcription (RT) was carried out with the iScript cDNA Synthesis Kit (Biorad, Hercules, CA, USA, 1708891) using 1 µg of total RNA. RT was performed in a thermal cycler as follows: 5 min at 25 °C for priming, 20 min at 46 °C for RT, and 1 min at 95 °C for RT inactivation.

Quantitative PCR reactions took place in a 384-well CFX384 Touch Real-Time PCR Detection System (Biorad), using 5 µL of iTaq Universal SYBR Green Supermix (Biorad) and 0.4 µM of forward and reverse primers in a final volume of 10 µL. Every reaction was performed in triplicate as follows: pre-heating 3 min at 95 °C, followed by 40 cycles of 10 s at 95 °C, 30 s at 60 °C, and for the melt curve, starting at 65 °C and increasing 0.5 °C every 5 s until reaching 95 °C.

Primers for *SLC19A1*, as well as the reference genes *RPS11* and *TPT1*, were based on the ones presented by Lee et al. [12]. Primers for cytosolic adenosine kinase, alternately referred to as ADK short (*ADK-S*) were designed in our laboratory based on the sequence NM_001123.4 from GenBank (NCBI). The primer sequences are:
*SLC19A1*Forward: CCTCGTGTGCTACCTTTGCTT
Reverse: TGATCTCGTTCGTGACCTGC*RPS11*Forward: CCGAGACTATCTGCACTACATCC
Reverse: GTGCCGGCAGCCTTG*TPT1*Forward: CACCTGCAGGAAACAAGTTTC
Reverse: GTCACACCATCCTCACGGTAG*ADK-S*Forward: TAGAGCATCGGACGCGGGCG
Reverse: GACTGACGTCATGGCTTCG

The result obtained for HEK293T cells maintained with 2200 nM FA was used as a control and results obtained with the other cell lines and/or conditions were compared to this value and expressed as arbitrary units (AU).

### 4.9. Statistics

All data are presented as mean ± SEM. Statistical analysis was performed using the GraphPad Prism 8.0.1 program. One-way ANOVA, followed by an uncorrected Fisher’s LSD test, was used when more than 2 groups were compared. Two-way ANOVA, followed by an uncorrected Fisher’s LSD, was used when comparing more than 2 groups and different conditions within each group. If the data had normal distribution, an unpaired two-tailed Student’s *t* test was used when only 2 groups of data were concerned. For linear correlation, the Pearson correlation coefficient was calculated with a confidence interval of 95%. *p* < 0.05 was considered statistically significant.

## 5. Conclusions

This study shows that several human cell lines (Jurkat, HEK293T, A549, Thermo Fisher) accumulate significant levels of ZMP at physiological levels of FA. Additional studies are required to assess the downstream targets of ZMP and whether this nucleotide, or its derivatives, could be used as a tumoral marker. It has been reported that cancer cells preferentially use the mitochondrial FA pool to synthesize formate, which in turn is converted to 10-formyl-THF, to sustain nucleotide synthesis, but at physiological levels of FA, it seems that certain cell lines (expressing low levels of the folate transporter SLC19A1) also use the cytosolic pathway, thus bypassing the mitochondrial dependence for nucleotide biosynthesis. We found a very good correlation (r = 0.94) between expression of SLC19A1 and accumulation of ZMP at physiological levels of folic acid. Indeed, the total content of purines was well correlated with SLC19A1 expression in cells maintained with high folic acid (r = 0.89). These results suggest an increased de novo purine synthesis in the cells expressing high levels of SLC19A1. The enzymes that regulate the de novo purine biosynthetic pathway form a macromolecular complex in the mitochondria: the purinosome. It makes sense that folic acid availability and the preferential use of de novo or salvage pathways would be regulated in some way. We hypothesize that there must be cellular mechanisms connecting the expression of SLC19A1 and a preferential use of the de novo versus the salvage pathway for purine biosynthesis.

## Figures and Tables

**Figure 1 ijms-24-12573-f001:**
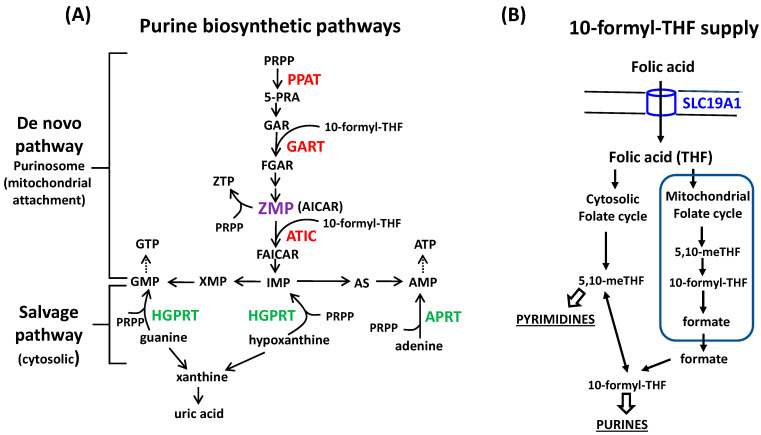
Schematic diagrams of the metabolic pathways studied. (**A**) Scheme of the de novo and salvage pathways of purine biosynthesis. Cancer cells can use both pathways to maintain their high purine requirements. An increased de novo pathway with a limited availability of folic acid, in the form of 10-formyl-THF (which mediates the transfer of 1C units), produces accumulation of 5′-aminoimidazole-4-carboxamide ribonucleotide (ZMP, also named AICAR), an intermediary of the de novo pathway. (**B**) Supply of 10-formyl-THF for the de novo purine biosynthesis. Folic acid is transported into the cells by the reduced folate carrier (RFC) SLC19A1, converted to tetrahydrofolate (THF), and subsequently to 10-formyl-TFH through multiple reactions taking place in the cytosol and/or the mitochondria. To simplify, the amino acids necessary for the de novo purine biosynthesis or in the folate-mediated one-carbon metabolism do not appear in the figures. Abbreviations not appearing in the text: 5-PRA: 5-phosphoribosylamine, 5,10-meTHF: 5,10-methylenetetrahydrofolate. Arrows indicate direct reactions and dotted arrows indicate that several reactions are involved.

**Figure 2 ijms-24-12573-f002:**
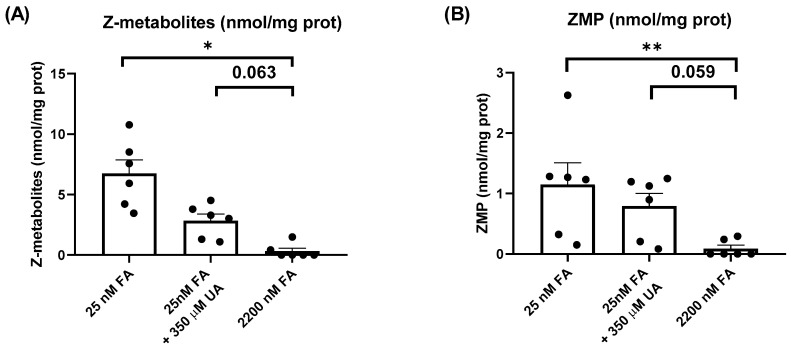
Physiological levels of folic acid induce ZMP accumulation in Jurkat cells. (**A**) Jurkat cells were incubated for 5 days in RPMI medium containing 25 nM folic acid (FA), 25 nM FA plus 350 µM uric acid (UA), or 2200 nM FA. Cell extracts were obtained with 0.4 N PCA and Z-metabolites were determined by using the Bratton-Marshall test. The graphs represent the mean ± SEM of 6 independent experiments, expressing the results as nmol Z-metabolites/mg protein. * *p* < 0.05. One-way ANOVA, uncorrected Fisher’s LSD test. (**B**) Jurkat cells were treated with different media, as described in (A), and ZMP levels were determined by HPLC in cell extracts. The graphs represent the mean ± SEM of 6 independent experiments, expressing the results as nmol ZMP/mg protein. ** *p* < 0.01. One-way ANOVA, uncorrected Fisher’s LSD test.

**Figure 3 ijms-24-12573-f003:**
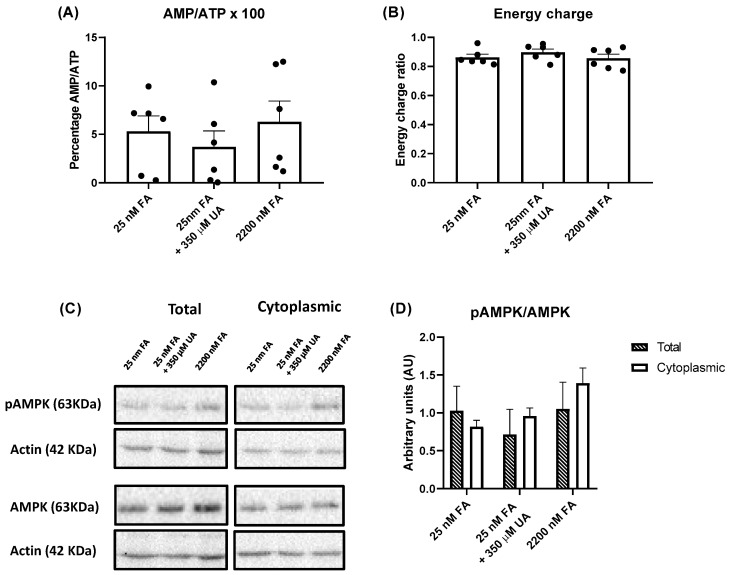
Physiological levels of folic acid do not induce AMPK activation in Jurkat cells. Jurkat cells were incubated for 5 days in RPMI medium containing 25 nM folic acid (FA), 25 nM FA plus 350 µM uric acid (UA), or 2200 nM FA. Cell extracts were obtained with 0.4 N PCA and purine levels were determined by HPLC. (**A**) AMP/ATP ratio, expressed as a percentage and represented as the mean ± SEM of 6 independent experiments. One-way ANOVA, uncorrected Fisher’s LSD test. (**B**) Cellular energy charge calculated as the ratio ([ATP] + 1/2 [ADP])/([ATP] + [ADP] + [AMP]) and represented as the mean ± SEM of 6 independent experiments. One-way ANOVA, uncorrected Fisher’s LSD test. (**C**) AMPK activity determined by Western blot. Jurkat cells were cultured in the above-mentioned conditions and whole-cell or cytoplasmic extracts obtained and analyzed by Western blot with antibodies against pAMPK (63 KDa), AMPK (63 kDa), and actin (42 KDa). (**D**) Quantification of AMPK activity. Bands obtained by Western blot were analyzed and quantified using the Image-J program. Each value obtained for pAMPK or AMPK was corrected by the corresponding levels of actin. The graph represents the mean ± SEM of 3 independent experiments. Total and cytoplasmic data were analyzed independently using one-way ANOVA, uncorrected Fisher’s LSD test.

**Figure 4 ijms-24-12573-f004:**
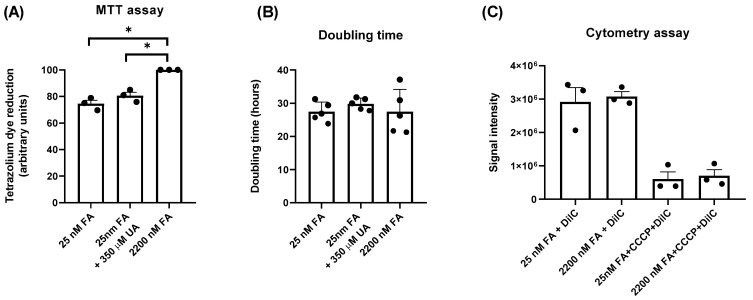
Physiological levels of folic acid decrease tetrazolium dye reduction (MTT assay) by Jurkat cells, without altering the mitochondrial membrane potential. (**A**) Jurkat cells were incubated for 5 days in RPMI medium containing 25 nM FA, 25 nM FA plus 350 µM UA, or 2200 nM FA and MTT assay was performed as described in Material and Methods. The graph represents the mean ± SEM of 3 independent experiments, using 2200 nM FA as a reference condition for normalization of the data and expressing the results as arbitrary units (AU). * *p* < 0.05. One-way ANOVA, uncorrected Fisher’s LSD test. (**B**) Jurkat cells were treated with different media, as described in (**A**), and doubling time was obtained using the formula described in Materials and Methods. This graph represents the mean ± SEM of 5 independent experiments. One-way ANOVA, uncorrected Fisher’s LSD test. (**C**) Jurkat cells cultured in RPMI medium with 25 nM FA or 2200 nM FA were incubated with DilC or DilC plus CCCP and signal intensity was quantified by cytometry. Results represent the mean ± SEM of 3 independent experiments. Unpaired *t*-test comparing 2200 nM vs. 25 nM FA with or without CCCP addition.

**Figure 5 ijms-24-12573-f005:**
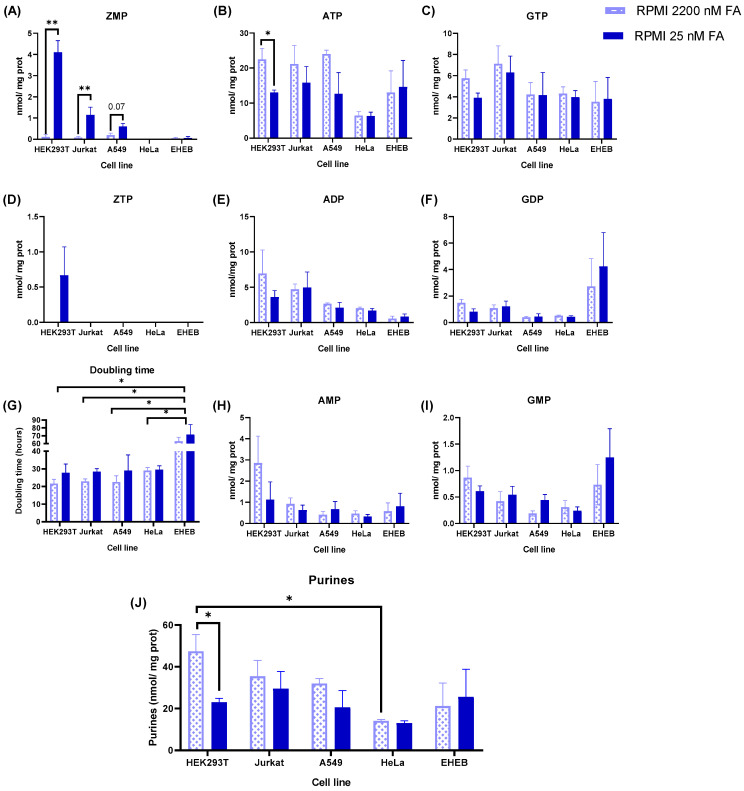
Physiological levels of folic acid induce ZMP accumulation in different tumoral cell lines. (**A**–**F**,**H**,**I**) HEK293T, Jurkat, A549, HeLa and EHEB cells were incubated for 5 days in RPMI containing 25 nM FA or 2200 nM FA. Cell extracts were obtained with 0.4 N PCA and nucleotides determined by HPLC. Results were expressed as nmol/mg protein. The graphs represent the mean ± SEM of at least 3 independent experiments, expressing the results as nmol nucleotide/mg protein. * *p* < 0.05; ** *p* < 0.01. Unpaired *t*-test comparing 2200 nM vs. 25 nM FA within each cell line. (**G**) Doubling time of the cells was obtained using the formula described in Materials and Methods. Unpaired *t*-test comparing 2200 nM vs. 25 nM FA within each cell line. One-way ANOVA, uncorrected Fisher’s LSD test comparing each condition between different cell lines. * *p* < 0.05. (**J**) Total purine concentration, calculated as the summatory of adenylates plus guanylates, in each cell line cultured with RPMI containing 2200 nM or 25 nM FA. Results were expressed as nmol/mg protein. Differences between cell lines were analyzed with a one-way ANOVA, uncorrected Fisher’s LSD test. Differences within cell lines, comparing 2200 nM vs. 25 nM folic acid, were analyzed with an unpaired *t*-test. * *p* < 0.05.

**Figure 6 ijms-24-12573-f006:**
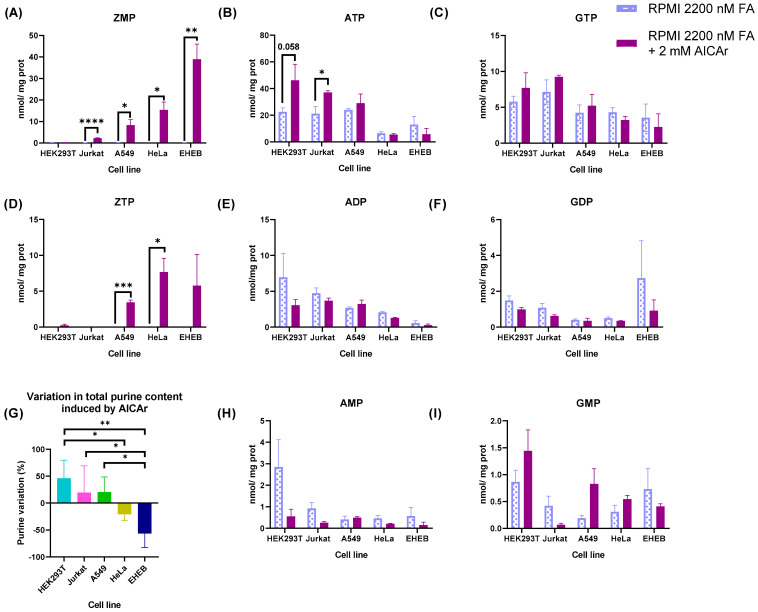
Purine alterations in different cell lines treated with 2 mM AICAr. HEK293T, Jurkat, A549, HeLa and EHEB cells were maintained for 4 days in RPMI containing 2200 nM FA and then incubated with the same medium with or without 2 mM AICAr for 24 h. (**A**–**F**,**H**,**I**) Cell extracts were obtained with 0.4 N PCA and purine levels determined by HPLC. Results were expressed as nmol/mg protein. The graphs represent the mean ± SEM of at least 3 independent experiments, expressing the results as nmol nucleotide/mg protein. * *p* < 0.05; ** *p* < 0.01; *** *p* < 0.001; **** *p* < 0.0001. Unpaired *t*-test comparing 2200 nM FA vs. 2200 nM FA + 2 mM AICAr for each cell line. (**G**) Variation of total purine content induced by AICAr. Total purine concentration for each cell line and condition was calculated as the summatory of adenylates plus guanylates. The graph represents the increase or decrease in total purines for each cell line treated with AICAr and compared with untreated cells. Results are expressed in percentages. * *p* < 0.05; ** *p* < 0.01. One-way ANOVA, uncorrected Fisher’s LSD test.

**Figure 7 ijms-24-12573-f007:**
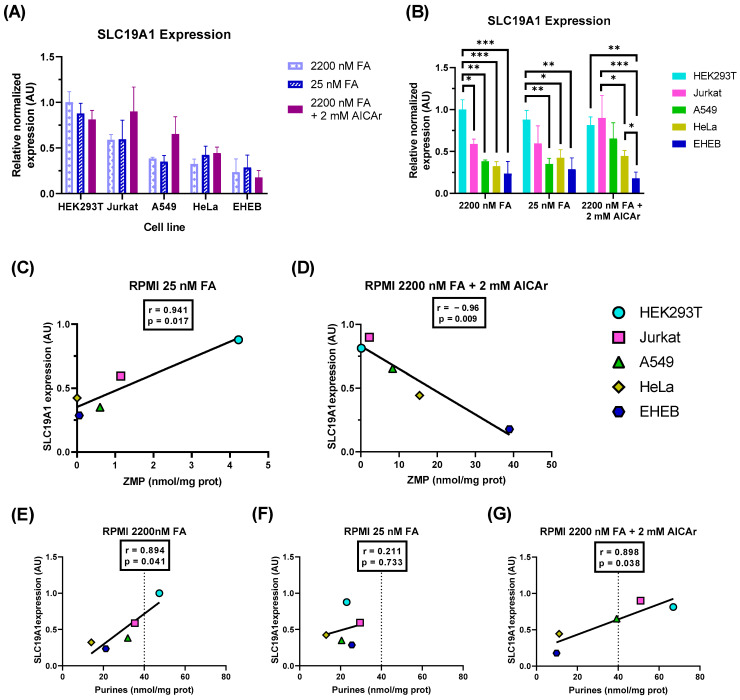
SLC19A1 mRNA expression in different cell lines. (**A**,**B**) HEK293T, Jurkat, A549, HeLa and EHEB cells were incubated for 5 days in RPMI containing 25 nM FA, 2200 nM FA, or 2200 nM FA + 2 mM AICAr (AICAr was added 24 h before collecting the cells). Total RNA was extracted and SLC19A1 mRNA levels quantified by qRT-PCR as described in Material and Methods. Results were expressed in arbitrary units (AU), using HEK293T cell line maintained with 2200 nM FA as a reference condition for normalization of the data, and grouped per cell line studied (**A**) or per condition assayed (**B**). The graphs represent the mean ± SEM of 3 independent experiments. One-way ANOVA (**A**) and two-way ANOVA (**B**), uncorrected Fisher’s LSD test. * *p* < 0.05; ** *p* < 0.01; *** *p* < 0.001. (**C**,**D**) Correlation between ZMP levels (in nmol/mg protein) and SLC19A1 expression (AU) in cells treated with 25 nM FA (**C**) or 2200 nM FA + 2 mM AICAr (D). (**E**–**G**) Correlation between total purine content, calculated as the summatory of adenylates and guanylates (in nmol/mg protein) and SLC19A1 expression (AU) in cells treated with 2200 nM FA (**E**), 25 nM FA (**F**), or 2200 nM FA + 2 mM AICAr (**G**). r = Pearson correlation coefficient.

**Table 1 ijms-24-12573-t001:** Purine nucleotide levels of Jurkat cells incubated with different media.

	Condition	25 nM FA	25 nM FA + 350 µM UA	2200 nM FA
Nucleotide		Mean ± SEM	Mean ± SEM	Mean ± SEM
ATP		15.8 ± 4.6	10.6 ± 1.3	23.0 ± 5.9
ADP		5.0 ± 2.2	1.8 ± 0.2	5.2 ± 0.7
AMP		0.6 ± 0.2	0.3 ± 0.1	1.1 ± 0.3
GTP		6.3 ± 1.6	5.3 ± 0.9	7.7 ± 1.9
GDP		1.2 ± 0.4	0.6 ± 0.1	1.2 ± 0.3
GMP		0.5 ± 0.2	0.3 ± 0.1	0.5 ± 0.2
ZMP		1.2 ± 0.4 **	0.8 ± 0.2	0.1 ± 01

Jurkat cells were incubated for 5 days in RPMI medium containing 25 nM folic acid (FA), 25 nM FA plus 350 µM uric acid (UA), or 2200 nM FA. Cell extracts were obtained with 0.4 N PCA and purine levels determined by HPLC. Results are the mean ± SEM of 6 independent experiments and are expressed as nmol/mg protein. ATP > ADP > AMP and GTP > GDP > GMP ratios were conserved in every experimental condition studied. ZMP was significantly increased in 25 nM FA compared with 2200 nM FA. ** *p* < 0.01. One-way ANOVA, uncorrected Fisher’s LSD test.

## Data Availability

Not applicable.

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
