# Peer review of "Purine Nucleotide Alterations in Tumoral Cell Lines Maintained with Physiological Levels of Folic Acid"

_ijms, 2023, doi:10.3390/ijms241612573_

Round 1

Reviewer 1 Report (New Reviewer)

The authors present an interesting manuscript on the effect of folate availability on the rate of purine synthesis. I have some observation listed below:

1)      The abstract should be rewritten to better underline the major points of interest of the manuscript, the present version is somehow confused.

 Accumulation of ZMP has been described not only in Lesh-Nyhan patients but also in patient with a hyperactivity of PRPP synthase, in fact in both cases a strong increase of the novo synthesis was observed. In the case of Lesh-Nyhan patients the increase of ematic and urinary urate is due to a 20 times increase of the rate of the novo synthesis which is far beyond the necessity to supply at the lack of the salvage pathway (HPRT). In fact, it has been demonstrated that the increase of the novo synthesis is occurring in fibroblast of the patient as a consequence of an increase of purinosome formation.

Fu R, Sutcliffe D, Zhao H, Huang X, Schretlen DJ, Benkovic S, Jinnah HA. Clinical severity in Lesch-Nyhan disease: the role of residual enzyme and compensatory pathways. Mol Genet Metab. 2015 Jan;114(1):55-61. doi: 10.1016/j.ymgme.2014.11.001. Epub 2014 Nov 8. PMID: 25481104; PMCID: PMC4277921.

The link between HGPRT deficiency and purinosome formation is still waiting to be elucidated. Therefore, I think that you could describe the cancer cells grown at physiological folate concentration as a model for cells HGPRT - only if you demonstrate that the low concentration of folate impact on purinosome formation, which by the way is possible. 

2)      Table 1. There is a mistake in the third column you write 25 nM FA + 350 μM FA instead of 25 nM FA + 350 μM UA.

3)      2-4 The intracellular nucleotide content is due not only to the rate of the synthesis but also to the catabolism and recycling. Therefore, even if there are no difference in the nucleotide contents in different kind of cells the lack of ZMP accumulation might be due to a different rate of the synthesis and/or purinosome formation

4)      2-5 The results in this paragraph needs to be better described because it is very hard to follow. In the presence of high concentration of AICAR and folate you are measuring the maximal rate of the final steps of de novo synthesis, in all the paper you completely forgot the activity of nucleotidases that play a role in the accumulation of purine compound and of their precursors.

5)      As for the abstract, the discussion must be simplified because is very hard to follow. You should point on the novelty of your findings more than on a superficial similarity of cancer cells with fibroblasts of Lesh-Nyhan patients.

Author Response

Reviewer 2 Report (Previous Reviewer 3)

the authors have responded to most of my previous criticisms.  However, they should adjust the numbers in table 1 and use only significant digits.  For example the ATP crow should be 16+/-5, 11+/-1, and 23+/-6.

Author Response

Reviewer 3 Report (New Reviewer)

In this study, the authors performed measurements of purine nucleotides and certain purine nucleotide precursors in a set of cancer cell lines maintained either on the medium with low folic acid (25 nm) or standard folic acid concentration (2200 nM). Apparently, this study was inspired by the 2020 PNAS paper (Lopez et al) which used essentially the same research design to investigate primary skin fibroblasts from individuals with Lesch-Nyhan disease. Overall, the results of the present study just support the role of folate in the de novo purine  biosynthesis, a textbook knowledge.  Of note, the claim that 25 nM folic acid is the physiological folate concentration does not really provide much justification for the study. Obviously, this claim is based on levels of folate measured in human plasma, but this raises several concerns. Thus, folic acid is not a physiological folate and is eventually converted to 5-methyl-THF, which is the main folate in plasma. More importantly though, folate levels in plasma should not be automatically translated to the cell culture: it is well known that cancer cells require abundant folate supply and typically, many cell lines do not grow well in low folate media. Further, cancer cell lines demonstrate different degree of dependence on the content of folate in media. Such diversity is linked to many parameters including levels of folate transporters, the ability to retain folate, levels of certain folate enzymes, and the overall folate utilization for metabolic purposes. The present study as well demonstrated cell-type specific responses leaving this reviewer with the question: what is the importance of presented data? It looks like the authors attempted to address this question by the statement that Jurkat cells can be used as a model to study alterations involved in LND. This does not sound like a very appealing idea: why not to use just LND fibroblasts for the same purpose? To summarize this reviewer’s opinion, the present study is purely descriptive with very limited novelty.

Round 2

Reviewer 1 Report (New Reviewer)

I believe that the manuscript is now more clear and better explained

Author Response

Thank you for your comments and the acceptance of the manuscript.

Jose

This manuscript is a resubmission of an earlier submission. The following is a list of the peer review reports and author responses from that submission.

Round 1

Reviewer 1 Report

See attached file.

Reviewer 2 Report

The authors of the article has identified intersting phenomena - accumulation of purine intermediate /e within the cancer cell lines when growing in the media with "physiological concentration" of folic acid.

I have few observations and questions.

I agree, that the observed effects relate to the cell in vitro cultivations, however, what is  physiological relevance of the given research? What is typical folic acid concentrations within the human body (blood, tissues) where malignacies develops? Moreover, how would you interpret, that diferent cancer cell lines are sensitive or not to "physiological folic acid concentrations".

in Figure 3 (G), please provide western blot on any housekeeping (GAPDH or actin etc.) protein expression.

Reviewer 3 Report

In this paper the authors have studied the effect of physiological folate concentrations on purine nucleotide levels in cells.  The authors draw many conclusions that are not supported by the results.

 Comments

 1.         Throughout the manuscript the authors describe alteration in “purine” levels.  I believe it would be more appropriate to refer to alterations in “purine nucleotides” in place of “purine”.  For instance, the title should read “Purine nucleotide alterations….” Instead of “Purine alterations…”.

 2.         In the abstract line 28 the authors indicate that the results in high folate “indicating that these cells preferentially rely on the salvage pathway for purine biosynthesis”.  I believe that “suggesting” would be a better choice of word here.  It is not clear to me why this result suggests that the cells preferentially rely on the salvage pathway.  Are there other possible conclusions?  It seems to me that this result could simply mean that at low concentrations of folate that the enzyme ATIC has become rate limiting in the production of purine nucleotides.

 3.         In line 26 of the abstract the authors state that GTP pools are increased.  However, Figure 6c does not show that GTP levels are significantly increased.  Also, ATP in A549 cells is not significantly increased.  This problem occurs throughout the manuscript.  If the results are not significant, then you cannot claim that there was a difference in the two treatment groups.  The text should be changed to adequately describe the results throughout the manuscript.

             Line 211, the authors state that the doubling time was “slightly increased” (4b).  I would conclude from the data that there was not difference in the results.

             Line 241 and 242, same problem ATP decreased but not significant.

             Lines 261 and 262, same problem.

             In line 274 the authors state that total purine content was decreased in 25 nM FA treated cells (in three cell lines), but they do not show statistics between 25 and 2500 nM FA in each of the cell lines.  They only show that at 2500 nM FA HEK293T were significantly different that HeLa cells.

             Lines 298 and 299, same problem.

             Lines 343 and 344, same problem.

             Because of the small changes many of which are not significant, I am not convinced of the conclusions drawn in lines 273 to 280.

 4.         In figure 2 there is significant variability in the results with 25 nM FA.  Why is that?

 5.         In table 1 only significant figures should be presented.  For instance, ATP levels at 25 nM FA should be presented as 1700 +/- 400.

 6.         ZMP is phosphorylated by Adenosine Kinase (line 286).  It would be interesting to know the levels of the adenosine kinase in the various cell lines.  Does adenosine kinase levels in the cells correlate with the result of figure 6.

 7.         Lies 416 and 417, “EHEB cells do not proliferate very much” is awkward.  I would just say that the doubling time of EHEB cells is much longer than the other cells.

 8.         Lines 422 and 423, the authors conclude that HeLa and EHEB cells rely more on the salvage pathway and that the other cells rely more on de novo synthesis.  It is not clear to me that the results suggest such a conclusion.  What is the concentration of purines in the growth media?  Are the purine concentrations in the growth medium enough to maintain the growth of these cells? 

 Lines 430 and 431, the authors provide the relative flux of de novo purine synthesis in the various cell lines.   The rate of de novo synthesis and purine salvage can be directly measured.  One could address this question in more unequivocal experiments.